# Electrocardiographic Changes in Liver Cirrhosis—Clues for Cirrhotic Cardiomyopathy

**DOI:** 10.3390/medicina56020068

**Published:** 2020-02-10

**Authors:** Letitia Toma, Adriana Mercan Stanciu, Anca Zgura, Nicolae Bacalbasa, Camelia Diaconu, Laura Iliescu

**Affiliations:** 1Department of Internal Medicine II, Fundeni Clinical Institute, 022328 Bucharest, Romania; letitia.toma@drd.umfcd.ro (L.T.); Adriana.mercanstanciu@drd.umfcd.ro (A.M.S.); 2Faculty of General Medicine, Carol Davila University of Medicine and Pharmacy, 022328 Bucharest, Romania; nicolaebacalbasa@gmail.com; 3Chemotherapy Department, OncoFort Hospital, 022328 Bucharest, Romania; medicanca@gmail.com; 4Department of Internal Medicine, Clinical Emergency Hospital of Bucharest, 022328 Bucharest, Romania; drcameliadiaconu@gmail.com

**Keywords:** cirrhosis, electrocardiography, ammonia, serum bilirubin, hypoalbuminemia

## Abstract

*Background and Objectives*: Cirrhotic cardiomyopathy is a chronic cardiac dysfunction associated with liver cirrhosis, in patients without previous heart disease, irrespective of the etiology of cirrhosis. Electrocardiography (ECG) is an important way to evaluate patients with cirrhosis and may reveal significant changes associated with liver disease. Our study aimed to evaluate ECG changes in patients with diagnosed liver cirrhosis and compare them to patients with chronic hepatitis. *Materials and Methods:* We evaluated laboratory findings and ECG tracings in 63 patients with cirrhosis and 54 patients with chronic hepatitis of viral etiology. The end points of the study were prolonged QT interval, QRS hypovoltage and T-peak-to-T-end decrease. We confirmed the diagnosis of cirrhotic cardiomyopathy using echocardiography data. *Results*: Advanced liver disease was associated with prolonged QT intervals. Also, QRS amplitude was lower in patients with decompensated cirrhosis than in patients with compensated liver disease. We found an accentuated deceleration of the T wave in patients with cirrhosis. These findings correlated to serum levels of albumin, cholesterol and ammonia. *Conclusions*: ECG changes in liver cirrhosis are frequently encountered and are important noninvasive markers for the presence of cirrhotic cardiomyopathy.

## 1. Introduction

Starting from 1953 [1], cirrhotic cardiomyopathy (CCM) has been defined as a separate entity in the spectrum of heart conditions, strictly related to liver damage. Since then, clinical and experimental trials have defined CCM as chronic cardiac dysfunction associated with liver cirrhosis, in patients without heart disease, irrespective of the etiology of cirrhosis [2]. From a physio-pathological perspective, CCM is characterized by a hyperdynamic state, with both diastolic and systolic ventricular dysfunction, prolonged ventricular depolarization and an inappropriate chronotropic response to stress [2,3,4]. CCM has also been described in pediatric populations, emphasizing the role of liver dysfunction in impaired cardiac function [5,6].

Several toxins have been implicated in cardiac dysfunction in cirrhosis. Excess of serum bile acids (encountered in liver failure due to decreased capacities of metabolism) has been shown to induce atrial arrhythmias (particularly atrial fibrillation) [7].

The prevalence of CCM is difficult to determine, mainly because the condition remains asymptomatic for a long time in the evolution of cirrhosis [8]. On the other hand, signs and symptoms of heart failure may resemble those of decompensated cirrhosis, making a differential diagnosis very difficult [9]. Furthermore, as the condition is under-diagnosed, clear management and treatment guidelines are unavailable.

A consensus had been reached in 2005 in establishing diagnostic criteria for CCM [10]. As such, these criteria included measurement of systolic dysfunction (a left ventricle ejection fraction of less than 55% or an impaired increase of cardiac output as a response to stress) and estimations of diastolic dysfunction (early diastolic atrial filling ratio, with a fraction E/A less than 1, prolonged deceleration time (TDE) over 200 ms and prolonged isovolumetric relaxation time over 80 ms). In addition, the consensus described supportive electrophysiological (prolonged QT interval, abnormal chronotropic response), biological (increased brain natriuretic peptide and troponin I) and ultrasonographic criteria (enlarged left atrium, increased myocardial mass) for the diagnosis of CCM.

Recent studies suggest that right ventricular dysfunction [11], enlarged right atrium and higher systolic pulmonary arterial pressure [12] are important markers of CCM and should be introduced as diagnostic criteria. Also, tissue strain imaging appears to be more sensitive in the evaluation of systolic dysfunction and should be used in the diagnosis of CCM [13].

As a result, in 2019 the Cirrhotic Cardiomyopathy Consortium released new guidelines redefining CCM and the importance of diastolic dysfunction [9]. The current proposed criteria include assessment of systolic dysfunction (a left ventricle ejection fraction of less than 50%) and several signs of diastolic dysfunction (low septal e’ velocity, high E/e’ ratio, high indexed volume of left atrium and high velocity of tricuspid regurgitation), while ECG modifications, MRI aspects and biomarkers are considered to add supportive information for the diagnosis.

Electrocardiography (ECG) is the first indicator of CCM. The most commonly encountered ECG pattern in CCM is a prolonged QT interval [3]. This occurs in more than half of cirrhotic patients and may lead to ventricular arrhythmias and sudden death [14]. Possible explanatory mechanisms for prolonged QT interval include dysfunction of membrane potassium channels and a hyper-reactivity of the sympathetic–adrenergic discharges, causing down-regulation of beta-adrenergic receptors [15]. Low-voltage ECG is also frequent in cirrhotic patients and has been associated with a higher mortality risk in patients without prior cardiovascular diseases [16]. Newer ECG markers for arrhythmogenesis include the T-peak to T-end interval (Tpe) and the Tpe/QT ratio [17].

This paper aims to evaluate the ECG aspect and alterations in patients with diagnosed liver cirrhosis, and compare them to patients with chronic hepatitis. The rationale for this study is based on the fact that the ECG is widely available and easy to perform in all patients, and it can bring important information for the management of these patients.

## 2. Materials and Methods

This is an observational, retrospective study and was approved by the Ethical Committee of Fundeni Clinical Institute, Bucharest, Romania (no 1638/ 17^th^ December 2017). Patients signed an informed consent form for the use of the data in scientific studies on admission in our clinic. Because of the retrospective design of the study and the lack of any intervention in the management of these patients during the study, no further consent was needed. Patient data were appropriately anonymized.

### 2.1. Patient Population

We analyzed the data of 167 patients with cirrhosis and 234 patients with chronic hepatitis (the control group) consecutively admitted to the Internal Medicine Clinic between January 2018 and July 2019.

The inclusion criteria were chronic liver disease associated with chronic viral infection, either hepatitis B (HBV), with or without association with hepatitis D (HDV) and hepatitis C (HCV). The positive diagnosis of infection was established by positive hepatitis B surface antigen and detectable HBV-DNA [18] or positive hepatitis C virus antibodies and detectable HCV viremia respectively [19]. Patients with chronic hepatitis C were selected from the pool of patients with indication for direct-acting antiviral treatment [20], prior to administration of the treatment.

Exclusion criteria applied were as follows:history of cardiovascular disease;antiarrhythmic treatment in the previous 6 months, including beta-blockers, or use of any other medication that can alter ECG patterns;diuretic treatment in the previous month;presence of pulmonary diseases (evaluated by history, clinical examination and chest radiography);presence of stage 3 or above chronic kidney disease or acute chronic kidney disease;presence of arrhythmias on ECG on admission;history of alcohol abuse;history of autoimmune liver disease;presence of hepatocellular carcinoma or other malignancies;anemia with hemoglobin levels under 10 g/dL;presence of thyroid disease (hypo or hyperthyroidism, autoimmune thyroiditis).

These criteria aimed to exclude causes for falsely altered ECG aspects, as dyselectrolytemias or increased inflammation markers may have an impact on liver function and cardiac electrophysiology.

As a result, the study population consisted of 63 patients with newly diagnosed liver cirrhosis of viral etiology and 54 patients with chronic viral hepatitis.

Demographic and medical data were collected from electronic source documents. The diagnosis of liver disease was based on clinical evaluation (cutaneous signs, hepato-splenomegaly), biological evaluation (liver function tests, ammonia levels, kidney function test, ionogram, serum bicarbonate blood cell count, coagulation parameters, virologic makers), noninvasive evaluation of liver stiffness by Fibroscan® and ultrasonography evaluation of the liver and portal mesenteric venous system. In patients with highly suggestive imaging aspects or increased tumor markers, CT scans were performed to exclude the presence of malignancy. Liver biopsy was not routinely performed. The diagnosis of cirrhosis was based of ultrasonography aspects and noninvasive evaluation of fibrosis, according to European Association for the Study of the Liver (EASL) guidelines [21,22]. Patients with cirrhosis were classified according to Child–Pugh scores. Decompensated cirrhosis was considered Child Pugh Class B and C [21,22]. Presence or history of cirrhosis complications (encephalopathy, ascites, variceal bleeding) were recorded.

### 2.2. Electrocardiography Analysis

A 12-lead ECG was performed for all patients on admission, on a calibrated device using a paper speed of 25 mm/s and standard amplitude of 10 mm /mV. The ECG tracings were separately assessed by a member of the study team, and manual measurements and calculations were performed and recorded. The following parameters were evaluated:QRS amplitude in all limb (DI, DII, DIII, aVL, aVF and aVR) and precordial (V1-V6) leads. Criteria for low-voltage QRS was amplitude of less than 0.5 mV in one of the limb leads and less than 1 mV in one of the precordial leads [23]. Mean values for QRS voltage in the limb leads and QRS voltage in the precordial leads were also calculated;QT interval, measured in leads DII and V6, from the beginning of the QRS interval to the end of the T wave. The lead with the longest QT interval was then considered, and the average QT interval from three consecutive heartbeats was recorded. Corrected QT (QTc) was calculated using Bazzet’s formula:
QTc= QT/√RR.(1)QT prolongation was defined as a corrected length interval of over 440 ms in male patients and over 460 ms in female patients;Tpe was calculated in all the leads using the tangent method, and a mean value was recorded. The peak was measured at the highest amplitude of the T wave relative to the isoelectric line. The T-end was defined as the intersection of the downslope of the T wave with the isoelectric line [24]. If there was a U wave present, the end of the T wave was considered at the lowest point between the T and the U waves.

### 2.3. Echocardiography Confirmation

In order to confirm/contradict the presence of CCM, we selected echocardiography data from current or previous analyses of the patients within a two-month time limit. Data selected were left ventricle ejection fraction (LVEF) estimated by Simpson biplan method, septal e’ velocity and E/e’ ratio.

### 2.4. Statistical Analysis

Data were evaluated using statistical software SPSS 18.0 (SPSS Inc., Chicago, IL, USA). The primary endpoints for the study were the evaluation of electrocardiographic changes, estimation of QT prolongation and Tpe shortening. Numerical variables were expressed as mean +/− standard deviation, and ANOVA test was used to compare between groups, with statistically significant *p*-values of less than 0.05.

## 3. Results

### 3.1. Baseline Characteristics

Overall, 63 patients with different stages of cirrhosis and 54 patients with chronic viral hepatitis were included in the trial. A total of 47.86% of patients were males (31 patients in the cirrhosis group and 25 patients in the hepatitis group). Table 1 presents the baseline characteristics of the two groups. For better characterization, patients with cirrhosis were further divided into subgroups according to Child–Pugh classification.

The results showed a significantly older age in patients with cirrhosis compared to patients with chronic hepatitis, as a marker of prolonged viral infection. There was no statistically significant difference in the etiology of liver disease, whether chronic HBV or HCV infection. Sodium levels were decreased in patients with cirrhosis, correlated with the severity of liver disease. Otherwise, we found normal potassium levels in all patients and a tendency of decreased magnesium, calcium and phosphorus levels along the progression of liver disease, without statistical significance. None of the patients presented acidosis. As expected, albumin levels, bilirubin, ammonia and international standard ratio (INR) also correlated well with the severity of liver damage. Notably, total cholesterol levels were associated with severe liver disease, as a marker of decreased liver synthesis.

We chose specific echocardiography markers of systolic and diastolic dysfunction to establish the diagnosis of CCM in our patients. LVEF did not reach statistical significance in correlation with the degree of liver dysfunction, while e’ velocity and E/e’ ratio were better correlated, suggesting a prevalence of diastolic dysfunction in our patients.

Complications of cirrhosis were found in 28 out of 63 patients. None of the patients presented acute portal vein thrombosis or digestive hemorrhage on admission. As expected, most of the patients with Child C cirrhosis presented encephalopathy and/or ascites in different degrees. Overall, 28 patients had encephalopathy: stage 1 was found in 2 patients with Child B and 3 patients with Child C, stage 2 was found in 5 patients with Child B and 10 patients with Child C and stage 3 was found in 8 patients with Child C. Ascites was found in 25 patients as follows: mild ascites in 1 patient with Child A, 5 patients with Child B and 4 patients with Child C, moderate ascites in 4 patients with Child A and 5 patients with Child C and severe or refractory ascites in 6 patients with Child C.

### 3.2. ECG Characteristics

ECG patterns were analyzed and compared between study groups (Table 2).

We found statistically significant differences between patients with chronic hepatitis compared to those with cirrhosis: mean QRS voltage was lower and more patients fulfilled the criteria for low voltage QRS. Furthermore, more patients had prolonged QT and shortened Tpe as markers of autonomic dysfunction and repolarization abnormalities.

A comparison between QTc and Tpe in patients with chronic hepatitis versus patients with decompensated cirrhosis (Child Pugh class B and C) proved that low albumin levels and increased bilirubin and ammonia levels correlated with QT duration (*p =* 0.03, *p =* 0.02 and *p* < 0.001 respectively) and shortening of Tpe (*p =* 0.04, *p =* 0.05 and *p =* 0.01 respectively). A linear regression pattern was demonstrated in the analysis of albumin, bilirubin and ammonia in patients with decompensated cirrhosis correlating with QTc and Tpe (Figure 1a–c, Figure 2a–c).

## 4. Discussion

Cirrhotic cardiomyopathy is a clinical entity with serious impact on the evolution and prognosis of patients with liver disease [10]. Since it is usually under-diagnosed, the management of these patients is based on management of cirrhosis and its complications alone [2]. While the condition remains asymptomatic in a resting state, due to decreased afterload (reduces vascular peripheral resistance), under stressful conditions (infections, exercise, hemorrhage) it can promote overt heart failure [25]. ECG changes are frequently encountered in patients with cirrhosis and were a part of the diagnostic criteria for CCM [3,10].

Current guidelines refer to ECG modifications as adjuvant criteria in the diagnosis of CCM [9]. The reason for this is the better in-depth characterization of the systolic, and especially the diastolic heart dysfunction, in cirrhosis using a multitude of parameters and novel ultrasonography techniques such as speckle tracking. The reasons for excluding ECG changes from the main diagnostic criteria of CCM are the limited number of studies and the inconsistent relationship to overall survival that these studies suggest. However, the guideline recognizes that prolonged QTc interval and short Tpe might be markers of CCM and poor prognosis of cirrhotic patients. ECG tracings are far more easily obtained than complex ultrasonography evaluations and can be used as a first assessment in cirrhotic patients with suspicion of CCM.

Studies suggest that QT prolongation can be found in more than 50% of patients with cirrhosis [26,27]. Furthermore, prolongation of QT interval has been associated with increased ventricular loading; both signs underline the typical pathophysiological aspect of CCM with increased cardiac output and ventricular overload [27]. Another trial found a prevalence of prolonged QT of up to 86% in patients with decompensated cirrhosis [28]; the results may have been biased by the fact that the most frequent etiology of cirrhosis in the study group was alcoholic hepatitis. In post-viral Child C cirrhosis, abnormal QTc intervals were found in about 60% of the patients, still without the possibility to exclude alcohol consumption as a risk factor [29].

The correlation between alcoholic liver disease and prolonged QT was the first described in the relationship between the heart and the liver [14]. Chronic alcohol consumption leads to progressive alcoholic liver disease but also myocardial dysfunction, with subclinical muscle injury and segmental delays in electric intra-cardiac conduction [30,31]. This defines alcoholic cardiomyopathy, with a high arrhythmogenic risk. On the other hand, electric abnormalities have been proven in nonalcoholic-related liver diseases, such as primary biliary cirrhosis, therefore strengthening that CCM is not purely related to alcohol abuse [32].

Several risk factors for prolonged QT in CCM have been described, including autonomic neuropathy, liver dysfunction (particularly Child score), dyselectrolytemias, left ventricle volume overload and administration of drugs with hepatic metabolism [17]. Autonomic nervous system impairment is of particular interest as an explanation for prolonged QTc in cirrhosis. Studies suggest that increased levels of norepinephrine correlate with QTc values [29]. Furthermore, the mechanism is opposite to that in normal subjects, where excess norepinephrine produces a lower QTc [33]; a clinical consequence of the autonomic dysfunction was the lack of increased heart rate despite high levels of norepinephrine, possibly due to down-regulation of beta-receptors.

In our study, we found that significantly more patients with cirrhosis had a higher QTc than patients with chronic hepatitis, with a prevalence of prolonged QT in the cirrhotic population of over 46%; the prevalence increased with the severity of liver disease estimated by Child–Pugh scores, in concordance with the results of Pourafkani et al [26]. We excluded patients with alcohol-related or inflammatory liver diseases in order to avoid association of these conditions to the ECG patterns discovered. C-reactive protein levels have been shown to predict prolonged QT intervals in patients with systemic lupus erythematosus [34], rheumatoid arthritis [35] and patients with other connective tissue diseases [36]. A possible explanation for the better correlation between QT interval and biologic parameters in our study could be the very restrictive selection of patients, excluding, at our best, many possible reasons for altered ECG traces.

QT prolongation is the pathophysiological basis for malignant arrhythmias and sudden death in cirrhosis, particularly following stressful situations, such as insertion of porto-systemic transjugular shunts or liver transplantation [37]. The direct correlation between QT and serum levels of brain natriuretic peptide suggests an underlying myopathy for this electro-physiologic abnormality. QTc may also be directly correlated to liver function parameters [38] and mean pressure in the portal vein [29]. In our study, we found a linear correlation between QTc and serum albumin, bilirubin and ammonia, with statistical significance, in patients with decompensated cirrhosis compared to the control group of chronic hepatitis. This may indicate further molecular mechanisms by which accumulation of toxins in liver failure may alter conduction of myocardial mechanisms. Hypoalbuminemia itself has been proven as an independent factor for the development of heart failure in elderly patients and patients with end-stage renal disease; a similarity may be found in patients with cirrhosis [39]. High levels of bilirubin have proven a protective effect on cardiovascular mortality by reduction of oxidative stress [40]. However, in our trial, high levels of bilirubin were correlated to ECG changes probably as an expression of the liver failure. Hyperammonemia promotes vasodilatation by nitric oxide signaling and endothelial dysfunction by the generation of reactive oxygen species [41], clearly showing its contribution to the hyperdynamic circulation in cirrhosis. In the skeletal muscle, ammonia decreases protein synthesis, promotes muscle autophagy and decreases adenosine-triphosphate synthesis, reducing contractility; these aspects may also apply to cardiac muscle.

Low-voltage QRS is rarely investigated in patients with cirrhosis, although it has been associated with increased mortality [16]. A recent trial [26] found that cirrhotic patients with low QRS voltages had higher MELD scores and lower albumin levels than those with normal ECG voltage; there was no difference in presence of ascites between the two groups. Potential causes for low-voltage ECG include pericardial effusion, peripheral edema, ascites and/or hypovolemia [23,42]. Unlike current literature data, we found that low-voltage QRS is statistically significant for cirrhosis patients. This discrepancy may be due to the fact that we compared the cirrhotic population with the noncirrhotic population. Patients with chronic hepatitis present normal serum albumin levels, as well as low MELD scores and normal sodium concentrations. These are protective factors for the development of peripheral edema, ascites and portal hypertension associated with microvoltage ECGs. A debated topic is whether or not the presence of ascites, per se, has an impact on ECG amplitude. A study with a prolonged follow-up of two patients with decompensated cirrhosis [23] proved that low-voltage ECGs were associated with fluid overload and normalized when diuretic treatment was administered; repeated paracentesis had no impact on QRS amplitude. On the contrary, another trial involving 20 patients with ascites of different etiologies found that the majority had low QRS amplitudes and attributed this to the fact that increased intra-abdominal pressure displaced the heart upwardly; the authors suggest that placing the ECG leads more cranially may partially correct this abnormality [43]. In our trial, the presence of ascites did not correlate directly to QRS voltage.

Tpe is a useful parameter in evaluating the transmural dispersion of repolarization [44] and a marker for ventricular arrhythmogenesis [45]. Prolonged Tpe intervals have been reported in patients with chronic hepatitis B [46] as indication of increased heterogenicity of ventricular repolarization. A recent analysis of Tpe in cirrhotic patients [47] found that shorter Tpe (less than 50 ms) is a prognostic factor for specific endpoints (death or liver transplantation). The authors found no significant variability between different stages of cirrhosis and QT duration or amplitude, or mean or maximum Tpe; this was explained by the fact that only a few of the patients included had decompensated cirrhosis (Child C). However, the fact that there was a direct association between minimum Tpe and negative outcome of the patients suggests that this is an important prognostic marker in CCM.

Our study has several limitations. First, the number of patients in each group was small, so strong recommendations for the general population cannot be issued. Second, the degree of dispersion of quantitative values was increased; however, statistical significance was achieved for a series of correlations that are currently absent in the literature. Third, the number of ECG parameters analyzed was small; this was derivative from the small study group, as we considered that taking into account a great number of determinations would divide the population into even smaller groups and would decrease the statistical significance.

## 5. Conclusions

Cirrhotic cardiomyopathy is a complication of cirrhosis that may appear early in the course of the disease and may remain clinically silent for many years. It is important to find specific diagnostic tools in order to accurately quantify the severity of the disease and its impact on the prognosis of patients. ECG changes appear early in the evolution of CCM, and follow-up of particular points of interest in association with standard monitoring parameters of liver cirrhosis may identify patients with higher risks. In this regard, QTc, QRS voltage and Tpe appear to be well correlated to the severity of the liver disease and may be used as diagnostic markers in the initial cardiologic evaluation of cirrhotic patients. 

## Figures and Tables

**Figure 1 medicina-56-00068-f001:**
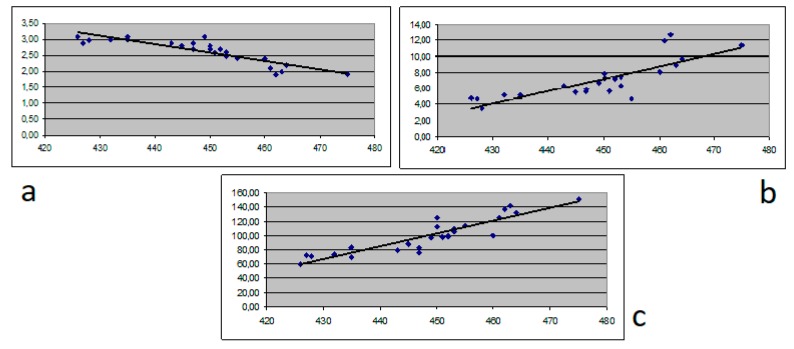
Correlation between albumin (**a**), bilirubin (**b**) and ammonia (**c**) and QTc.

**Figure 2 medicina-56-00068-f002:**
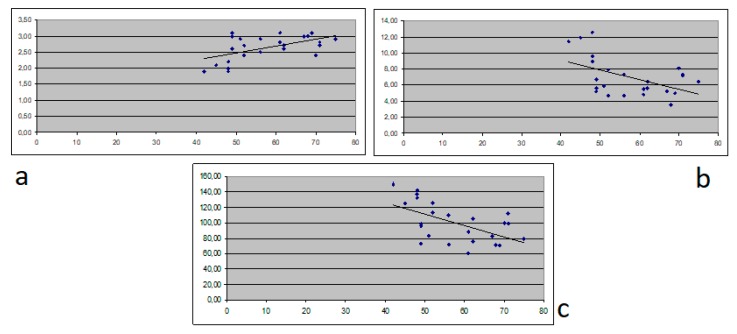
Correlation between albumin (**a**), bilirubin (**b**) and ammonia (**c**) and T-peak to T-end interval (Tpe).

**Table 1 medicina-56-00068-t001:** Baseline characteristics of the study groups.

Parameter	Chronic hepatitis (*N =* 54)	Cirrhosis Child A (*N =* 18)	Cirrhosis Child B (*N =* 20)	Cirrhosis Child C (*N =* 25)	*p* Value
Age (y)	48.4 +/− 5.3	50.4 +/− 10.5	55.7 +/− 13.2	60.8 +/− 14.3	**0.0043**
Gender (M/F)	25/29	9/9	11/9	11/14	0.67
Etiology (HBV/HCV)	28/26	8/10	7/13	9/14	0.589
ALT (IU/mL)	46 +/− 28	62+/− 23	75 +/− 39	77 +/− 45	0.52
AST (IU/mL)	78 +/− 45	104+/− 67	112 +/− 53	101 +/− 62	0.25
Total bilirubin (mg/dL)	0.5 +/− 0.4	1.2 +/− 0.7	2.5 +/− 1.3	7.1 +/− 3.8	**0.035**
Albumin (g/dL)	4.2 +/− 0.6	3.3 +/− 0.3	3.1 +/− 0.7	2.8 +/− 0.9	**0.028**
Total cholesterol (mg/dL)	212 +/− 39	177 +/− 48	152 +/− 46	134 +/− 72	**0.045**
Ammonia (μg/dL)	32 +/− 17	47 +/− 12	71 +/− 23	89 +/− 21	**0.021**
Creatinine (mg/dL)	0.9 +/− 0.3	0.9 +/− 0.4	1.0 +/− 0.3	0.9 +/− 0.4	0.626
Sodium (mEq/L)	143 +/− 3	140 +/− 4	136 +/− 4	131 +/− 6	**0.05**
Potassium (mEq/L)	4.4 +/− 0.5	4.2+/− 0.6	4.3 +/ 0.5	4.1 +/− 0.7	0.274
Total calcium (mg/dL)	9.4 +/− 0.8	9.2 +/− 0.7	9.3+/− 0.6	9.0 +/− 0.7	0.053
Magnesium (mg/dL)	2.1 +/− 0.5	2.1+/− 0.7	2.0 +/− 0.6	1.9 +/− 1.7	0.62
Phosphorus (mg/dL)	3.6 +/− 1.2	3.2 +/− 0.7	3.3 +/− 0.4	3.0 +/− 0.8	0.09
Bicarbonate (mmol/L)	24 +/− 3	25 +/− 4	23 +/− 4	23 +/− 5	0.13
INR	0.9 +/− 0.1	1.1 +/0.3	1.3 +/− 0.2	2.2 +/− 1,5	**0.0012**
LVEF (%)	62 +/− 9	55 +/− 6	52 +/− 7	52 +/− 3	0.07
Septal e’ velocity (cm/s)	12 +/− 3	11 +/− 3	8 +/− 2	6 +/− 2	**0.01**
E/e’ ratio	19 +/− 3	16 +/− 4	13 +/− 4	11 +/− 2	**0.001**

IU international units, AST aspartate aminotransferase, ALT alanine aminotransferase, INR International Standardized Ratio, LVEF left ventricle ejection fraction. Bolded values are statistically significant. HBV: hepatitis B virus; HCV: hepatitis C virus.

**Table 2 medicina-56-00068-t002:** ECG characteristics and comparison between patients with chronic hepatitis and patients with cirrhosis.

Parameter		Chronic Hepatitis (*N =* 54)	Cirrhosis Child A (*N =* 18)	Cirrhosis Child B (*N =* 20)	Cirrhosis Child C (*N =* 25)	*p* Value
**Mean QRS voltage**	**Precordial Leads**	10.2 +/− 3.5	9.8 +/− 2.5	9.1 +/− 3.1	8.9 +/− 2.9	0.043
	**Limb Leads**	5.1 +/− 1.9	4.9 +/− 1.4	4.8 +/− 2.1	4.5 +/− 1.7	0.032
Criteria for QRS Hypovoltage		3 (5.5%)	6 (33.33%)	12 (60%)	16 (64%)	0.041
Heart Rate (bpm)		81 +/− 19	82 +/− 21	88 +/− 16	92 +/− 15	0.039
QRS Duration (ms)		85 +/ 16	85 +/− 14	83 +/− 12	84 +/− 17	0.134
QTc (ms)		418 +/− 1	445 +/− 27	451 +/− 23	459 +/− 31	0.045
Criteria for Prolonged QT		2 (3,7%)	6 (33.33%)	10 (50%)	13 (52%)	0.021
Mean Tpe (ms)		73 +/− 18	71 +/− 17	70 +/− 23	64 +/− 15	0.023
Criteria for Shortened Tpe (<50ms)		1 (1.8%)	3 (16.66%)	9 (45%)	15 (60%)	0.034

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
