# Peer review of "Electrocardiographic Changes in Liver Cirrhosis—Clues for Cirrhotic Cardiomyopathy"

_1010-660X, 2020, doi:10.3390/medicina56020068_

Round 1

Reviewer 1 Report

I read with great interest the original manuscript titled: Electrocardiographic changes in liver cirrhosis - clues for cirrhotic cardiomyopathy by Toma L et al. The authors describe electrocardiographic disturbance in CCM . The study is interesting because it reintroduces rhythm disturbance in cirrhosis – a criterion which was used in 2005 consensus guidelines, but removed in the 2019 adult guidelines (Izzy et al). This omission, points for and against the omission will make this paper interesting to readers.

The paper has the potential to re-ignite the topic of electrocardiographic disturbance in cirrhosis. However, to do so certain things need to be addressed:

What was the reason to choose chronic hepatitis as a control? Can the authors use age matched normal adults as controls?

What was the magnesium, phosphorous and acid base status of the patients ? these abnormalities can affect rhythm by itself.

Rainer et al correlated bile acid levels with atrial dysrhythmias. What were the bile acid levels?

Outcomes measures (mortality/hospital LOS/etc.) are needed to re-establish the importance of rhythm abnormalities that have been pointed out in this article. Otherwise this will be an “incidental” finding with no consequence.

The new guidelines do not use electrocardiographic disturbance as a criterion in CCM. This criterion was used in 2005 guidelines. It has also been shown that prolonged QTc increases odds of dying pre-LT in adults (bernardi et al) and children (Arikan et al). This paper also suggests that rhythm abnormality is seen in cirrhosis. Why the criteria have been dropped from the new guidelines should be addressed in discussion, Points in favor of re-including the abnormality in the definition should be mentioned. This will make the discussion interesting and will strengthen this paper

Introduction sections should include New guidelines (Izzy et al 2019). Rhythm issues have been implicated in cirrhosis and children with liver failure by Rainer et al (heart) and Arikan et al. these papers need to be mentioned.

In discussion please describe what the authors speculate the mechanisms for electrocardiographic abnormalities

Author Response

Thank you very much for your comments and suggestions, we tried to respect them as good as possbie. We think that they greatly improved the quality of our paper. This is our point-by point answer: 

Point raised: What was the reason to choose chronic hepatitis as a control? Can the authors use age matched normal adults as controls?

Answer: We chose chronic hepatitis patients as control patients because chronic hepatitis can also determine ECG modifications; as such, the modifications caused by the viral infection would no represent a sourse of bias in the evaluation of cirrhotic patients. If we had compared infected cirrhotic patients with normal control we would no be able to evaluate which modifications came from the infection and which from the impaired liver function.  

Point raised: What was the magnesium, phosphorous and acid base status of the patients ? these abnormalities can affect rhythm by itself.

Answer: We inserted serum levels of magnesium, phosphorus and bicarbonate in the table containing basic data obtained from the study. The values were not statistically significand between study groups.

Point raised: Rainer et al correlated bile acid levels with atrial dysrhythmias. What were the bile acid levels?

Answer: Unfortunately, we do not routinely determine bile acid levels, so this information could not be obtained from the patient record to be used in the study.

Point raised: Outcomes measures (mortality/hospital LOS/etc.) are needed to re-establish the importance of rhythm abnormalities that have been pointed out in this article. Otherwise this will be an “incidental” finding with no consequence.

Answer: First of all, this study is based on recent data from patients with cirrhosis and chronic hepatitis (data obtained during the interval January 2018- July 2019). Patients with chronic hepatitis have a life expenctany of over 2 years, as would be the maximum follow-up period for this particular cohort. Secondly, we chose patients with advanced liver disease without prior specific treatment, such as diuretics or beta-blockers. After their initial evaluation (included in this study), the patients received optimal medication, which can affect both ECG pattern and overall survival. As a result, evaluating survival in our cohort would bring no real value.

Point raised: The new guidelines do not use electrocardiographic disturbance as a criterion in CCM. This criterion was used in 2005 guidelines. It has also been shown that prolonged QTc increases odds of dying pre-LT in adults (bernardi et al) and children (Arikan et al). This paper also suggests that rhythm abnormality is seen in cirrhosis. Why the criteria have been dropped from the new guidelines should be addressed in discussion, Points in favor of re-including the abnormality in the definition should be mentioned. This will make the discussion interesting and will strengthen this paper

Answer: Comments on the new guidelines have been inserted in the discussion section.

Point raised: Introduction sections should include New guidelines (Izzy et al 2019). Rhythm issues have been implicated in cirrhosis and children with liver failure by Rainer et al (heart) and Arikan et al. these papers need to be mentioned.

Answer: The proper citations have been inserted in the introduction and discussion sections.

Point raised: In discussion please describe what the authors speculate the mechanisms for electrocardiographic abnormalities 

Answer: Possibile mechanisms for ECG abnormalities have been discussed.

Reviewer 2 Report

Thank you for the opportunity to review the paper "ECG changes in liver cirrhosis". I have the following comments: 

1- In abstract results mostly unrelated to the aim of the study. Second phrase of conclusion can not be drawn from current study. I don't think much can be said about cirrhotic cardiomyopathy from current study as there was no echocardiography performed and including that in the title of study is misleading. Aim is poorly defined and 

2- introduction: I dont see the reason cirrhotic cardiomyopathy is being discussed. There is no echocardiographic data presented and it is not clear if cirrhotic patients had cirrhotic cardiomyopathy

3- methods: criteria used for chronic hepatitis vs. cirrhosis are not clear. Not clear how ECG measurements were performed (software or manual). The T peak to T end interval not correctly noted and described. Not clear why control group is  drawn from chronic hepatitis patients not healthy individuals particularly since there is no liver biopsy and chronic hep also could associate with ECG changes. Criteria for decomoensated cirrhosis not defined. QRS hypovoltage needs to be changed to low-voltage and the definition of it does not seem to be correct.

Why are tumor markers mentioned in method section for identifying cirrhotic patients

4- results: disorganized. Exclusion criteria not to be mentioned here ( hepatorenal)

5- discussion also has redundant sections. Most parts of conclusion can not be drawn from this study

Overall study has flaws which makes results of low interest for readership

Author Response

Thank you very much for your comments and suggestions, we tried to respect them as good as possible. This is our point-by point answer:

In abstract results mostly unrelated to the aim of the study. Second phrase of conclusion can not be drawn from current study. I don't think much can be said about cirrhotic cardiomyopathy from current study as there was no echocardiography performed and including that in the title of study is misleading. Aim is poorly defined and 

Answer: We included some echocardiographic data to prove the existence of cardiomyopathy in our patients and validate the conclusions. Our aim is to focus on ECG chances which we believe can help establish a quick suspicion if not confirmed diagnosis of CCM. Echocardiography and special techniques such as strain evaluation are not widely available and can be more time and resource consuming than a simple mandatory ECG.

introduction: I dont see the reason cirrhotic cardiomyopathy is being discussed. There is no echocardiographic data presented and it is not clear if cirrhotic patients had cirrhotic cardiomyopathy

Answer: We included several echocardiography data in Table 1, to prove the existence of CCM.

methods: criteria used for chronic hepatitis vs. cirrhosis are not clear. Not clear how ECG measurements were performed (software or manual). The T peak to T end interval not correctly noted and described. Not clear why control group is  drawn from chronic hepatitis patients not healthy individuals particularly since there is no liver biopsy and chronic hep also could associate with ECG changes. Criteria for decomoensated cirrhosis not defined. QRS hypovoltage needs to be changed to low-voltage and the definition of it does not seem to be correct.

Answer: We defined chronic hepatitis versus cirrhosis according to EASL guidelines. Criteria for decompensated cirrhosis have been added. Better description of ECG measurements has been added. T peak to T end interval and QRS amplitude have been re-defined. We chose not to use healthy controls as there could have been no way to distinguish between ECG changes due to chronic viral infection and ECG changes due to liver failure.

Why are tumor markers mentioned in method section for identifying cirrhotic patients

Answer: Tumor markers are mentioned (as well as CT scans) as useful for excluding patients with malignancies (as mentioned in the exclusion criteria)

results: disorganized. Exclusion criteria not to be mentioned here ( hepatorenal)

Answer: The results has been improved, adding comments on electrolyte balance and the ultrasonography aspects that supported the diagnosis of CCM. The sentence regarding hepatorenal syndrome has been removed.

discussion also has redundant sections. Most parts of conclusion can not be drawn from this study

Answer: We have revised the discussion section and added new data regarding the lates guidelines and possible explanatory physio-pathological mechanisms for our findings.